# Microbiota, a New Playground for the Omega-3 Polyunsaturated Fatty Acids in Cardiovascular Diseases

**DOI:** 10.3390/md19020054

**Published:** 2021-01-23

**Authors:** Guy Rousseau

**Affiliations:** 1Centre de Biomédecine, Hôpital du Sacré-Cœur, CIUSSS du Nord de l’île de Montréal, 5400 boul. Gouin Ouest, Montréal, QC H4J 1C5, Canada; Guy.Rousseau@umontreal.ca; Tel.: +(1)-514-338-2222 (ext. 3421); 2Département de Pharmacologie et de Physiologie, Université de Montréal, Montréal, QC H3T 1J4, Canada

**Keywords:** Omega-3 PUFA, cardiovascular diseases, microbiota, dysbiosis

## Abstract

Several cardioprotective mechanisms attributed to Omega-3 polyunsaturated fatty acids (PUFAs) have been studied and widely documented. However, in recent years, studies have supported the concept that the intestinal microbiota can play a much larger role than we had anticipated. Microbiota could contribute to several pathologies, including cardiovascular diseases. Indeed, an imbalance in the microbiota has often been reported in patients with cardiovascular disease and produces low-level inflammation. This inflammation contributes to, more or less, long-term development of cardiovascular diseases. It can also worsen the symptoms and the consequences of these pathologies. According to some studies, omega-3 PUFAs in the diet could restore this imbalance and mitigate its harmful effects on cardiovascular diseases. Many mechanisms are involved and included: (1) a reduction of bacteria producing trimethylamine (TMA); (2) an increase in bacteria producing butyrate, which has anti-inflammatory properties; and (3) a decrease in the production of pro-inflammatory cytokines. Additionally, omega-3 PUFAs would help maintain better integrity in the intestinal barrier, thereby preventing the translocation of intestinal contents into circulation. This review will summarize the effects of omega-3 PUFAs on gut micro-biota and the potential impact on cardiac health.

## 1. Cardioprotective roles of omega-3 PUFAs

Omega-3 polyunsaturated fatty acids (PUFAs) are often associated with cardio-protective effects. Recent studies suggest that omega-3 PUFAs could attenuate the deleterious impact of a dysbiosis by acting on gut microbiota, a common characteristic observed in patients with cardiovascular diseases. In this review, we will discuss the potential role of omega-3 PUFAs on the microbiota and their possible effects on cardiovascular diseases.

Large controlled randomized trials comprising more than 32,000 participants provided evidence of a major reduction of cardiovascular events by omega-3 PUFA supplements [1,2,3]. However, these beneficial effects are not reported universally. For example, the OMEGA trial indicated that the effects of 1 g/day of omega-3-acid ethyl esters-90 did not further reduce sudden cardiac death and other clinical events [4]. Probably one of the most important analyses of omega-3 PUFA supplementation in cardiovascular disease involving 68,680 patients indicates that omega-3 PUFA supplementation is not associated with a lower risk of all-cause mortality, cardiac death, sudden death, myocardial infarction (MI), and stroke [5]. The results of the ORIGIN study (12,536 high-risk patients with or at risk of diabetes) reveal that omega-3 PUFA treatment does not reduce the total mortality, cardiovascular mortality or any cardiovascular events compared to the placebo [6]. The difference between beneficial and no effect studies are still a matter of debate, but the dose or the ratio between the different omega-3 PUFA could be an issue. More recently, the addition of 4 g/day of icosapent ethyl demonstrated a reduction of the ischemic events in patients statin-treated high-risk patients [7], suggesting that a higher dosage or a unique omega-3 PUFA could be more effective. The latter point has been observed in our experimental study, demonstrating that the addition of eicosapentaenoic acid (EPA) and docosahexaenoic acid (DHA) in the diet did not afford protection in contrast to each omega-3 PUFA alone at the same dose [8].

In experimental studies, results indicated a reduction of infarct size in many species fed with a multiple long-chain omega 3 PUFA [9,10,11] or with only one [12,13]. Overall, it seems that there is a large consensus in favor of reduced infarct size with omega-3 PUFA in animal experiments.

Until the present day, there were a lot of mechanisms by which omega-3 PUFA could exert cardio-protective effects. The anti-inflammatory properties of omega-3 PUFA are certainly the first one that we have considered. These properties could be related to their incorporation in cell membrane phospholipids, largely at the expense of AA (omega-6 PUFA), which is pro-inflammatory [14]. Other possibilities have been uncovered with the identification of G-protein coupled receptors (GPCR) that interact with fatty acids (GPR43, GPR120) [15,16]. For instance, DHA interacts with GPR120 and could inhibit iΚB kinase as well as the production of pro-inflammatory cytokines, such as tumor necrosis factor-alpha (TNFα). DHA and EPA may also inhibit NF-κB activity by the interaction with PPARγ or interference with early events before NF-κB activation [17].

We also observed that a high omega-3 PUFA diet is cardio-protective via a mechanism involving Akt activation [11], an enzyme identified to be part of the key biochemical pathway component the reperfusion injury salvage kinase (RISK) [18]. When activated at the onset of the reperfusion, these kinases confer cardio-protection by mPTP opening inhibition [19]. DHA could also inhibit the opening of the mPTP and result in a reduction of infarct size [8] by an unknown mechanism.

In addition to the direct effect of omega-3 PUFA, we must also consider the metabolites involved in the resolution phase of inflammation known as resolvins (Rv). We observed that RvD1 administration before the onset of reperfusion reduced myocardial infarct size in a porcine model [20]. We also observed that when there is inhibition of the main enzymes involved in DHA transformation to RvD1 (COX-2 and 15-LOX), plasma RvD1 concentrations are reduced and the cardio-protection is abolished [21]. Similarly, Keyes et al. reported that RvE1 administration in a rat model of MI significantly reduces infarct size and increases Akt and Erk activity [22]. This indicates the potential role of these metabolites in the cardio-protection observed with omega-3 PUFAs.

Another cardio-protective effect that omega-3 PUFAs and their metabolites could induce, but that is still speculative at this moment, is the impact of the composition of the microbiota. As we will see, omega-3 PUFAs can positively alter gut microbiota and preserve an intestinal function by reducing this deleterious contribution to cardiovascular diseases.

## 2. Microbiota and Dysbiosis

The human gut microflora comprises over 1000 species and more than 7000 strains [23], representing 10^13^–10^14^ bacterial cells, which is ten times more numerous than other cells. Healthy gut microbiota is mainly composed of the phyla *Firmicutes* and *Bacteroidetes*, representing around 90% of the human gut flora, followed by *Actinobacteria*, *Verrucomicrobia*, and *Proteobacteria*. The large intestine hosts over 70% of all microbes in the human body. Pathogens, such as *Campylobacter jejuni*, *Salmonella enterica*, *Vibrio cholera*, and *Escherichia coli* can also be found, but in low numbers. While microbiota composition varies between healthy individuals in terms of different taxa proportions and rapid bacterial alterations are observed in humans, their magnitude is modest [24].

Dysbiosis is frequently observed in cardiovascular diseases patients [25,26]. Dysbiosis is an “imbalance” in the gut microbial community. This imbalance could be due to the gain or loss of community members or changes in the relative abundance of microbes. In addition to obesity [27], metabolic syndrome [28], and type 2 diabetes [29], evidence suggest that diet can also induce dysbiosis [30], which is often associated with an increase in inflammation [31,32].

Different diets can induce dysbiosis, whose reported changes in the microbiota differ from diet to diet. For example, a diet rich in complex carbohydrates increases *Bifidobacteria* [33], while a high-fat and high sugar diet results in an increase in *Clostridium innocuum*, *Catenibacterium mitsuokai*, and *Enterococcus* spp. [34]. Elevated diet in fat or carbohydrate in humans is associated with a decrease in *Bacteroidetes* and an increase in *Firmicutes* [35], which is also reported in animals [36,37]. In humans, an omega-3-PUFA rich diet (600 mg daily) for 14 days increased the abundance of several bacteria producing the short-chain fatty acid (SCFA) butyrate [38], known to have anti-inflammatory effects [39,40]. A 14 day washout reverses these effects. Similarly, in mice, fish oil treatment for 15 days elicits significant gut microbiota changes. This result indicates that diets rapidly affect microbiota [41]. A high-fat diet causes dysbiosis by increasing the ratio of Gram^−^/Gram^+^ bacteria [42]. The increase in Gram^−^ bacteria, of which γ-*Proteobacteria* is a part, correlates with an increase in lipopolysaccharide (LPS), which promotes inflammation. In turn, this inflammation contributes to a modification of the microbiota, favoring the proliferation of other bacteria of an inflammatory nature [43]. This, therefore, suggests that a diet rich in ω-3 would be beneficial for health and that it helps restore the balance of the microbiota (Figure 1).

Along with nutritional influences, studies have shown that sex [44,45,46,47,48,49] also modulates microbiota composition. For example, in the same experimental conditions, males generally show more microbiota changes than females [47]. According to a recent study, male mice are more sensitive to DHA than females, exhibiting more significant changes in their gut microbiota [44]. However, some studies indicate that adding two omega-3 PUFAs (EPA and DHA) induced significant changes in female mice microbiota [50], while in other studies, males and females showed similar variations [41]. These data underscore the importance of continuing studies on the impact of sex to better understand the potential sex-specific mechanisms underlying microbiota’s influence on health.

The gut microbiota participates in several physiological functions for maintaining health, including its involvement in the catabolic pathways to produce SCFA and cometabolites such as ammonia, phenols, indoles, various amines and thiols [51]. The gut microbiota also contributes to regulating the intestinal mucosal barrier, the control of nutrient uptake and metabolism, the maturation of immunologic tissues, and the prevention of the propagation of pathogenic microorganisms [52,53,54,55,56]. Thus, the regulation of these functions is essential to limit diseases, including cardiovascular diseases.

### 2.1. Myocardial Infarction and Microbiota

No direct evidence is presently available that demonstrates gut microbes influences on infarct size [57]. However, a mango-flavored Goodbelly juice, containing a probiotic, L. Plantarum 299 v, reduced the size of the myocardial infarct [58]. It is difficult to confirm that the positive effect is only related to the probiotic since other components are present. A study by Lam et al. indicates a link between gut microbiota metabolites and the severity of the myocardial infarction [59]. In another study, the data indicate that the microbiota’s richness is higher in the rat MI group than in the sham group at day seven after the onset of ischemia. This change parallels intestinal barrier impairment documented by reducing the occludin (a tight junction protein) and the Chiu pathological scores of mucosal injuries [60].

Interestingly, we also observed that MI induces a change in intestinal barrier integrity 14 days post-MI in our rat ischemia/reperfusion model [61]. Overall, data suggest a link between the gut microbiota and MI involving alteration of the gut barrier integrity.

### 2.2. Microbiota and Cardiovascular Diseases

Additionally, the evidence is accumulating for a role of the microbiota in the development of heart failure. According to the “gut hypothesis”, the decrease in cardiac output and the systemic congestion observed in this condition would promote intestinal mucosal ischemia and/or edema, as well as contributing to the translocation of bacteria, an increase in circulating endotoxins, thereby contributing to inflammation [62,63]. In patients with heart failure, an increase in intestinal permeability was also noted compared to controls [64].

While inflammation is well described in heart failure, several clinical studies hypothesized the benefit of reduced inflammation in heart failure have yielded rather disappointing results [65,66,67,68]. However, a recent study has revived this solution by targeting more patients with or without heart failure following a myocardial infarction benefiting from targeted anti-cytokine therapy [69].

Studies have shown that gut microbiota may also modulate the risk factors involved in developing MI and heart failure, such as atherosclerosis [70], hypertension [71], and obesity [72]. For instance, in *Apoe*^−/−^ mice, treatment with ampicillin was used to decrease the number of bacteria, reduced low-density lipoprotein, very-low-density-lipoprotein cholesterol levels, and the atherosclerotic aortic lesion, compared with controls [73]. Gut microbiota was also reported to produce TMA, a precursor of trimethylamine-N-oxide (TMAO), which promotes atherosclerotic plaques [74].

Microbiota from *spontaneously hypertensive rats* (SHRs) transferred to Wistar Kyoto (WKY) rats results in a significant systolic blood pressure increase (26 mm Hg) compared with controls [75]. Others have reported that the *Firmicutes* to *Bacteroidetes* ratio is only increased in SHRs compared with pre-hypertensive SHRs or healthy rats [71], suggesting a link between microbiota composition and blood pressure.

Gut microbiota isolated from obese mice and transferred to germ-free recipients resulted in a 20% increase of total body fat than the transfer of gut microbiota isolated from lean mice [76], supporting a link between microbiota and obesity.

## 3. Potential Roles of Microbiota

The microbiota dysbiosis has already numerous impacts on cardiovascular diseases and could be involved at different levels, leading to increased myocardial damage. The following observations urge us to determine the impact of a “dysbiotic diet” on the ischemic myocardium.

The intestinal barrier regulates the absorption of nutrients, electrolytes, and water from the lumen. It prevents the passage of pathogenic microorganisms (or of their products) and toxic substances into the bloodstream [77]. Different features protect the barrier’s integrity, including a mucus layer and a monolayer of epithelial cells interconnected by tight junctions. The mucus layer contains immunoglobulin A and antimicrobial peptides that facilitate gastrointestinal transport and protection against bacterial invasion.

The tight junctions consist of complex protein structures (e.g., claudin, occludin, and tricullin) that form mechanical links between epithelial cells [78]. Gut microbes or their metabolites may modulate the intestinal barrier integrity: the exact mechanisms are unclear, but may include the following: (1) intestinal epithelial cell renewal [79] and cell death [80]; (2) activation of signalling pathways involved in barrier integrity (phosphatase, kinase) [81]; and (3) production of metabolites that reduce inflammation [82].

Compromised intestinal barrier integrity correlates with chronic, low-grade inflammation [82]. The translocation of the bacterial lipopolysaccharide (LPS) component of Gram-negative bacteria in the bloodstream could play a significant role [83]. LPS concentrations could be 10–50 times lower than in septicemia or infection, but could still be sufficient to evoke inflammation [84].

Gut microbiota could also affect inflammation levels by acting on the vagus nerve. Vagus nerve stimulation reduces infarct size [85,86,87,88] by a mechanism involving cholinergic activation’s anti-inflammatory properties. Recent studies show that the microbiota’s composition could modulate the vagus nerve activation by an undefined mechanism [89,90,91].

Microbiota could also affect infarct size by producing SCFAs, through their anti-inflammatory properties [92,93]. Non-digested polysaccharides are fermented by gut microbes, generating SCFAs, mostly acetate, propionate, and butyrate [94,95]. These metabolites present well-characterized, anti-inflammatory properties, and modulate cellular functions through G-protein coupled receptors (GPR41, GPR43, and GPR109A) or inhibiting histone deacetylases [96,97,98,99].

The gut microbiota metabolizes trimethylamine (TMA)-containing compounds (choline, phosphatidylcholine, and carnitine) to TMA, a precursor of trimethylamine-N-oxide (TMAO) produced by hepatic cells [100]. Several publications have reported that TMAO heightens cardiovascular risk by promoting atherosclerotic lesions [74] or platelet activation [101]. TMAO suppresses reverse cholesterol transport [102] and up-regulates pro-atherogenic scavenger receptors [103].

Accumulating experimental data suggest the gut microbiota participates in different pathological states where inflammation is involved, which could impact infarct size. MI is an inflammatory pathology involving neutrophil accumulation [104] and the production of pro-inflammatory substances, such as cytokines [105,106] or arachidonic acid derivatives [107,108,109]. Infarct size reduction may also be caused by the attenuation of neutrophil accumulation [110], injection of anti-inflammatory molecules [111,112] or resolvins [20,21,22], which participate in the resolution phase of inflammation. These observations urge us to determine the impact of a “dysbiotic diet” on the ischemic myocardium.

Moreover, in addition to the impact on myocardial infarct size, microbiota could also interfere with mechanisms that alter the healing of infarcted myocardium and heart failure development.

TMAO and butyrate are among molecules released by the microbiota that are suspected of impairing myocardial healing or precipitated heart failure. Some even suggest that the TMAO would be a predictor of mortality in the case of heart failure. While the mechanism remains speculative, most authors agree that the inflammation associated with the presence of TMAO is linked to the increased production of pro-inflammatory cytokines such as TNFα, IL-6, and IL-1β [113] or the activation of NOD-, LRR- and pyrin domain-containing protein 3 (NLRP3) inflammasome and NF-κB. Also, TMAO could promote apoptosis [114]. This increase in the production of TMAO is associated with an augmentation of the bacteria that belong to the genus of *Clostridium*, *Escherichia*, and *Proteus* [115].

Others have reported that concentrations of butyrate in feces and plasma are lower in cases of heart failure [116]. An anti-inflammatory response to butyrate via NF-κB inhibition is reported in several in vitro and in vivo studies where a decrease in myeloperoxidase concentrations cyclooxygenase-2 and cytokines has been identified [117,118,119,120], with some effects related to the FFAR3 receptor. This reduction of butyrate is associated with a reduction in bacteria that belong to the genus of *Roseburia*, *Faecalibacterium*, and *Eubacterium* [121]. In these circumstances, a high omega-3 PUFA diet could attenuate dysbiosis as well as limit damage due to the microbiota.

In conclusion, the data suggest that a diet rich in omega-3 PUFAs affects the composition of the microbiota, thereby alleviating the dysbiosis seen in patients with cardiovascular disease and reducing the deleterious effects associated with dysbiotic microbiota. However, it remains essential to know whether this effect is observable, regardless of gender and whether age alters this relationship or not. Overall, this new avenue of interventions is exciting but further studies are needed to have a better understanding of omega-3 PUFAs on the modulation of microbiota and these effects on cardiovascular diseases.

## Figures and Tables

**Figure 1 marinedrugs-19-00054-f001:**
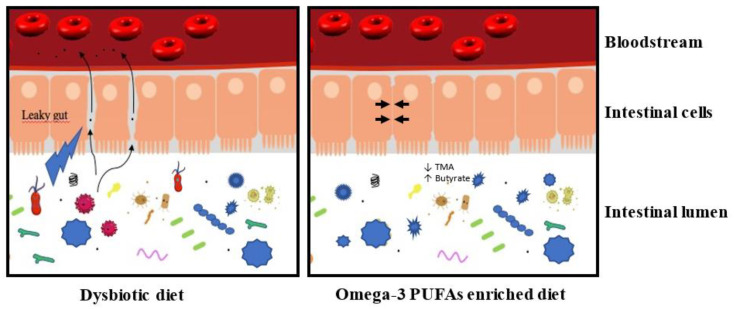
A dysbiotic diet disturbs the composition of the intestinal microbiota and affects the integrity of the intestinal barrier. In these conditions, the intestinal lumen’s content could transfer to the bloodstream and induce a low-grade inflammation. This low-grade inflammation contributes negatively to the size of the myocardial infarction and the development of heart failure. An enriched omega-3 PUFA diet could maintain a healthy microbiota and preserve the integrity of the intestinal barrier (→ ←). This microbiota is associated with a decrease in TMA production and an increase of butyrate (see text for details). In these conditions, the enriched omega-3 PUFA diet prevents the induction of low-grade inflammation.

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
