# Peer review of "Microbiota, a New Playground for the Omega-3 Polyunsaturated Fatty Acids in Cardiovascular Diseases"

_marinedrugs, 2021, doi:10.3390/md19020054_

Round 1
Reviewer 1 Report
The Topic of the submitted article fits very well into the general scope for the special Issue "Marine Functional Food Products" of "Marine Drugs"
This article is a review of the literature on the potential links between the intestinal microbiota and cardiovascular disease. The article is divided into 3 parts and the author makes the link between the protective power of omega 3 (PUFAs) and the intestinal microbiota. The PUFAs would allow a better balance of the intestinal flora leading to a reduction of the inflammatory risk which would be the cause of cardiovascular diseases. However, the author remains very cautious in his understanding of this potential mechanism.
The article is generally well written and easy to follow and understand for someone who is not a specialist. Some modifications are however necessary for reading fluency.
line 39: MI should be identified as Myocarditis Infection (MI) I think
line 48: what is EPA and DHA? Please define them
DHA is defined later but too late line 119 as docosahexaneoic acid.
Figure 1 should be more explicit: caption the different arrows present and the different players in the intestinal lumen.
After these minor modifications, I recommend the publication of the article "Microbiota, a new playground for the omega-3 polyunsaturated fat acids in cardiovascular diseases" in "Marine Drugs".
Author Response
line 39: MI should be identified as Myocarditis Infection (MI) I think
Answer: this has been corrected myocardial infarction (MI)
line 48: what is EPA and DHA? Please define them
Answer: This is now corrected. …eicosapentaenoic acid (EPA) and docosahexaenoic acid (DHA)
Figure 1 should be more explicit: caption the different arrows present and the different players in the intestinal lumen.
Answer: We have added explanations in the legend of figure 1.
Reviewer 2 Report
The manuscript by Rousseau provides a concise review of cardioprotective omega-3 polyunsaturated fatty acid (omega-3-PUFA) and its role in enteric dysbiosis that exacerbates inflammation-induced cardiovascular diseases including myocardial infarction, hypertension, and cardiac failure. Rousseau provided a broad spectrum of the role of omega-3-PUFA, microbiome populations and intestinal signaling pathways underlying barrier integrity. The field is emerging and, indeed, mini reviews may establish fundamental insights into anti-inflammatory metabolic profiles with therapeutic purposes. This review is well-written. Nevertheless, this review could be further strengthened by addressing comments below.
Comments
- Considering the nature of review articles, please consider reducing other extensive information and focus on the logistics whereby omega-3-PUFA regulates population of microbiota and enteric dysbiosis.
- Please make the abbreviation of omega-3-PUFA consistent.
- Despite well-written manuscript, I located several issues with typos. Additional proofreading may help in prior to the final submission.
Author Response
Considering the nature of review articles, please consider reducing other extensive information and focus on the logistics whereby omega-3-PUFA regulates population of microbiota and enteric dysbiosis.
Answer: We have reduced some sections of the article.
Please make the abbreviation of omega-3 PUFA consistent.
Answer: omega-3 PUFA is now used throughout the text.
Despite well-written manuscript, I located several issues with typos. Additional proofreading may help in prior to the final submission.
Answer: A native English editor has corrected the manuscript.